# VF-PS: How to Select Important Participants in Vertical Federated Learning, Efficiently and Securely?

**Jiawei Jiang**[1,2,3]    **Lukas Burkhalter**[2]    **Fangcheng Fu**[4*]    **Bolin Ding**[5]
**Bo Du**[1]    **Anwar Hithnawi**[2]    **Bo Li**[6]    **Ce Zhang**[2]

[1] School of Computer Science, Institute of Artificial Intelligence,
Hubei Key Laboratory of Multimedia and Network Communication Engineering,
National Engineering Research Center for Multimedia Software, Wuhan University
[2] ETH Zürich    [3] OceanBase, Ant Group    [4] Peking University    [5] Alibaba Group
[6] University of Illinois at Urbana–Champaign
{jiawei.jiang, dubo}@whu.edu.cn
{lukas.burkhalter, anwar.hithnawi, ce.zhang}@inf.ethz.ch
ccchengff@pku.edu.cn    bolin.ding@alibaba-inc.com    lbo@illinois.edu

## Abstract

Vertical Federated Learning (VFL), that trains federated models over vertically partitioned data, has emerged as an important learning paradigm. However, existing VFL methods are facing two challenges: (1) *scalability* when # participants grows to even modest scale and (2) *diminishing return* w.r.t. # participants: not all participants are equally important and many will not introduce quality improvement in a large consortium. Inspired by these two challenges, in this paper, we ask: *How can we select $l$ out of $m$ participants, where $l \ll m$, that are most important?*

We call this problem *Vertically Federated Participant Selection*, and model it with a principled mutual information-based view. Our first technical contribution is VF-MINE—a *Vertically Federated Mutual INformation Estimator*—that uses one of the most celebrated algorithms in database theory—Fagin's algorithm as a building block. Our second contribution is to further optimize VF-MINE to enable VF-PS, a group testing-based participant selection framework. We empirically show that vertically federated participation selection can be orders of magnitude faster than training a full-fledged VFL model, while being able to identify the most important subset of participants that often lead to a VFL model of similar quality.

## 1 Introduction

The quality of machine learning models heavily relies on the quality and volume of available data. As our understanding of machine learning shifts towards such a data-centric view, mechanisms of making data available to machine learning have attracted intensive interests [1, 2]. Towards this goal, Federated Learning [3–5] is an emerging research area that focuses on training a single ML model using all data available in a federated "*data consortium*" in a secure and privacy-preserving way.

***Vertical Federated Learning.*** In this paper, we focus on a specific federated learning scenario: *Vertical Federated Learning* (VFL) [5]. As illustrated in Figure 1(a), the VFL setting involves a data consortium that consists of $m$ participants $\mathcal{P} = \{P_1, ..., P_m\}$. Each participant holds *a disjoint subset of features* associated with the same entity (e.g., different hospitals holding medical records of the same patient but for different types of diseases). The goal of vertical federated learning is to train a single ML model over the joint feature space, in a secure and privacy-preserving way, without communicating their data to each other [6–10].

---

[*]Corresponding author.

36th Conference on Neural Information Processing Systems (NeurIPS 2022).

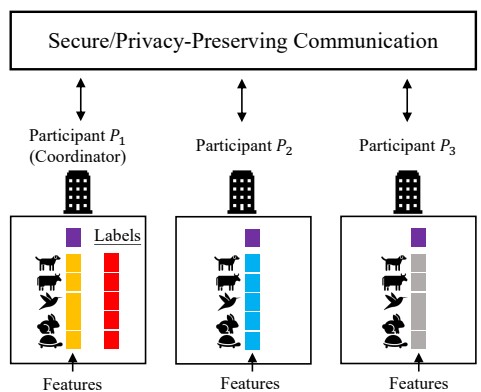

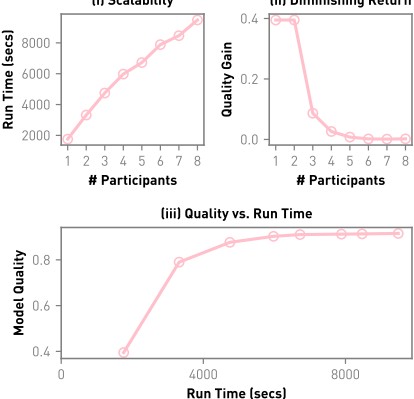

(a) In the Vertical Federated Learning (VFL) setting, each participant holds a disjoint set of features for the same set of entities; the coordinator also holds the ground truth labels. These participants communicate in a secure and privacy preserving way in order to train a single machine learning model over the joint feature space.

(b) VFL faces two challenges: (i) *Scalability* and (ii) *Diminishing Return* w.r.t # participants, since many participants in a large consortium might not provide significant quality gain. As a result, (iii) a dominating amount of the computation cost for a large consortium might not lead to any quality gain. (LR, Bank dataset)

Figure 1: Vertical federated learning and challenges

***Technical Challenges of VFL.*** Despite many recent efforts that significantly improved the scalability and efficiency of model training in the VFL setting, two great challenges linger, as illustrated in Figure 1(b). The first challenge is *scalability* w.r.t # participants. Because of the requirement on security and privacy, many VFL approaches have to resort to expensive methods such as Homomorphic Encryption [11, 12] and Secure Multi-party Computation [13, 14]. This greatly limits the scalability of today's VFL systems. Another challenge is *diminishing return*, also when # participants grows. Since not all participants contribute equally informative knowledge, it is not unusual to observe that adding more participants does not provide significant quality gain on the final ML model. These two challenges, when put together, form a quite "ironic" picture, as illustrated in Figure 1(b)(iii) — *supporting a large data consortium makes training time significantly longer, however, a dominating amount of these computations might not be translated into any significant quality gain.*

***Vertically Federated Participant Selection (VF-PS).*** Whereas designing more efficient model training methods in the VFL setting continues to be crucially important, in this paper, we focus on an orthogonal direction to accommodate these challenges. Given a data consortium with $m$ participants, instead of trying to speed-up the training of a single ML model over all $m$ participants, we ask: "*Can we identify $l \ll m$ participants that are most important, and only train expensive VFL models on them instead?*" We call this problem *Vertically Federated Participant Selection (VF-PS)*.

***From Feature Selection (FS) to VF-PS: Connections and New Challenges.*** At first glance, the VF-PS problem is closely connected to traditional feature selection problems [15, 16]. This is definitely true and is what, in our opinion, makes VF-PS promising — the last half-century of study on feature selection provides not only principled theoretical framework but also strong evidence that we should be able to significantly decrease # participants (# features) involved but still obtain similar quality. However, VF-PS imposes unique challenges compared to feature selection. VF-PS should (1) provide *no weaker* security and privacy guarantees compared with other VFL training systems [9] and (2) should be much *more efficient* than training a single VFL model over all participants.

***Summary of Technical Contributions.*** In this paper, we focus on a specific, while one of the most natural, way of modeling VF-PS: *we should select the subset of participants that jointly preserve the greatest amount of mutual information (MI) between features and labels.* Under this view, designing a VF-PS algorithm involves two technical issues: (1) *how to estimate MI for a subset of participants?* and (2) *how to search the subset of participants that preserve the greatest amount of MI?*

- **(C1)** Our first contribution is VF-MINE—a Vertically Federated Mutual INformation Estimator. Given a subset of participants, VF-MINE estimates MI between the features of these participants and the labels. VF-MINE protects data privacy via homomorphic encryption. The most surprisingly

technical observation is that one of the most popular MI estimators [17], used by `sklearn`, can be made orders of magnitude faster. This is by optimizing data movement between different participants, with Fagin's algorithm [18] as a building block.

- **(C2)** Our second contribution is VF-PS, that uses VF-MINE to search for the subset of participants that preserves the greatest amount of MI in a group testing-based framework. We provide a set of optimization techniques to further optimize data movement between different participants.
- **(C3)** We provide rigorous security analysis for the proposed methods. We then empirically show that, our VF-PS framework can be up to two orders of magnitude faster than training full-fledged VFL models. Moreover, VF-PS can often select a much smaller subset of participants but maintain a similar model quality.

## 2 Preliminaries

*Federated Learning.* Federated learning is a category of distributed learning technique that enables different organizations or users to collaboratively learn machine learning (ML) models, without exposing their personal data to other parties [19–21]. Assume $\mathcal{X}$ denotes instance features, $\mathcal{Y}$ denotes instance labels and $\mathcal{I}$ denotes instance IDs, federated learning can be classified into two categories, i.e., horizontal federated learning and vertical federated learning, according to how $(\mathcal{X}, \mathcal{Y}, \mathcal{I})$ are distributed across different participants.

*Privacy Protection Techniques.* Preserving the privacy of input data is a crucial requirement of many deployments of federated learning. Therefore, these systems are often used in conjunction with privacy protection techniques. In what follows, we briefly summarize key techniques used in this space. 1) *Differential Privacy* randomly perturbs the transferred data (e.g., gradient) with noises, such as Gaussian noise [22], Laplacian noise [23], and Binomial noise [24]. 2) *Homomorphic Encryption* encrypts all data before they are transferred [12, 25, 26] and supports arithmetic computation on encrypted data. 3) *Secure Multiparty Computation* (MPC) enables multiple parties to collaboratively compute an agreed-upon function without leaking local input to any other party except for what can be inferred from the output [13, 14, 27, 28].

## 3 Vertically Federated Participant Selection via Mutual Information

In this section, we describe our algorithm for Vertically Federated Participant Selection (VF-PS).

### 3.1 Vertically Federated Participant Selection

We assume a standard data model, following previous efforts in VFL [7, 8] — let there be $m$ participants and $N$ data instances, and $X \in \mathbb{R}^{N \times F}$ be the joint feature space where $F$ is the dimension of the joint feature space. In the VFL setting, this joint feature space $X$ is vertically partitioned over different participants — each participant $P_i \in \mathcal{P}$ holds a subset of features (columns) of $X$, denoted by $X_i$. Without loss of generality, we have: $X = [X_1 \, ... X_i ... \, X_m]$. Among all $m$, there is a *leader participant*, which holds the instance labels $Y$. We assume that all participants agree on the same identifier (ID) of each data instance.

*Security Requirements.* Following previous systems, e.g., Pivot [9], VF²Boost [29], we have the following standard security requirements. 1) *Feature Security*: Features $X_i$ on each participant cannot be shared with any other party. 2) *Aggregation Security*: Only the leader participant can obtain an *aggregated* result over all *participants'* local data. 3) *Identity Security*: No other party, except the participants, can obtain the identifiers (IDs) of instances.

*Mutual Information-based VF-PS.* Similar to the classic feature selection problem, we focus on one of the most natural ways to model VF-PS — we aim at selecting the *subset* of participants that jointly preserve the greatest amount of mutual information with the label. Formally, the computation that we hope to conduct, under the previous security requirements and vertically partitioned data model, is

$$\max_{\alpha_1, ..., \alpha_l} MI([X_{\alpha_1} \, ... \, X_{\alpha_l}]; Y); \; s.t, \forall i \in [l] : \alpha_i \in [m] \text{ and } \forall i, j \in [l] : \alpha_i \neq \alpha_j. \quad (1)$$

where $MI(\cdot)$ estimates the mutual information from finite samples, and $l < m$ is a pre-defined constant specifying the number of participants to select.

## 3.2 VF-MINE: A Vertically Federated Mutual Information Estimator

As we see in Equation 1, solving this problem involves two different questions: (1) *how to estimate MI given a fixed subset of participants, i.e., $MI([X_{\alpha_1} \ldots X_{\alpha_l}]; Y)$?* and (2) *how to efficiently search for the best subset, i.e., $\max_{\alpha_1,\ldots,\alpha_l}$, without enumerating exponential many possibilities?* We focus on the first problem in this section, and leave the second question to Section 3.3.

### 3.2.1 Baseline Method

*KNN-based MI Estimator.* We focus on a standard family of MI estimators that is widely used in practice (e.g., in `sklearn`'s feature selection module), which is based on $k$-nearest neighbors (KNN)[17, 30]. Let there be $N$ data pairs $\mathcal{D} = \{(x_j, y_j)\}$ where $y_j$ is a label and $x_j$ is a feature vector. A KNN-based MI estimator enumerates multiple *query* data pairs. For each query $q = (x_q, y_q)$:

1. We first compute $N_q$, the number of examples in $\mathcal{D}$ that share the same labels as $y_q$, i.e.,

$$N_q = |\{(x, y) \in D : y = y_q\}|$$

2. We then construct $NN_{k,y_q}$, containing all $k$-nearest neighbors among data examples that share the same label. Let $\bar{d}_q$ be the maximal distance between examples in $NN_{K,y_q}$ and $x_q$.
3. We then compute $m_q$, the number of examples in $\mathcal{D}$ that has a distance to $x_q$ smaller than $\bar{d}_q$.

Given $Q$ as a subset of query data pairs, we estimate the mutual information as follows: $\frac{1}{|Q|} \sum_q \left( \psi(N) - \psi(N_q) + \psi(K) - \psi(m_q) \right)$, where $\psi$ is the digamma function $\psi(x) = \frac{d}{dx} ln(\Gamma(x)) \sim lnx - \frac{1}{2x}$.

*Baseline Implementation of VF-MINE.* We now provide a baseline implementation of this estimator in the vertically federated setting. We assume that we use *Euclidean distance* as our distance metric, but our technique can be naturally applied to other distance metrics. We choose additive *homomorphic encryption* (HE) to protect data privacy since it is widely adopted in the literature [8, 11, 25]. We use $HE.Enc(\text{-})/HE.Dec(\text{-})$ to denote the encryption/decryption function, and [-] an encrypted data item. Below, we present a single VF-MINE routine. There are three roles in the system — *key server*, *aggregation server*, and *participant*:

- *Key server.* The key server generates public and private HE (homomorphic encryption) keys $(pk, sk)$, allocates $sk$ to the leader participant, and $pk$ to all participants and the aggregation server. We assume the key server is trusted and does not collude with any party.
- *Aggregation server.* An aggregation server provides an `addition` operator that securely aggregates multiple encrypted data items: $[\mathbf{d}] = HE.Sum(\{[\mathbf{d}_i]\}; pk)$.
- *Participant.* Each participant $P_i$ holds a feature subset of all $N$ data instances. For a query data pair $q = (x_q, y_q)$, each participant $P_i$ calculates $\mathbf{d}_i = [(x_{q,i} - x_{j,i})^2 \text{ for } (x_j, y_j) \in \mathcal{D}]$, the distances between its local features of $x_q$ and data pairs in $\mathcal{D}$. We call $\mathbf{d}_i$ the partial distances. Then, $P_i$ encrypts partial distances and sends $[\mathbf{d}_i] = HE.Enc(\mathbf{d}_i; pk)$ to the aggregation server via the `addition` operator. There is a *leader participant* that holds the labels. It receives the complete distances $[\mathbf{d}]$ from the aggregation server, decrypts it to $\mathbf{d} = HE.Dec([\mathbf{d}]; sk)$, and calculates MI, responding to the procedure of KNN-based estimator.

### 3.2.2 Fagin-Inspired Method

The baseline method has a potential performance issue, that is, it has to aggregate all the data instances. Since the operation through homomorphic encryption is often time-consuming, this baseline method can be inefficient in practice, especially for many large-scale datasets. To decrease the cost, a potential way is to decrease the number of aggregated instances. Motivated as such, our first question is — *can we aggregate fewer data instances and meanwhile obtain the correct results?* We observe that the vertical KNN task can be seen as aggregating multiple "sub-ranking"s into a "global ranking". Based on this interpretation, we propose to leverage a textbook aggregation algorithm, called *Fagin*, to efficiently find $k$-nearest neighbors.

*Fagin Algorithm.* The problem of top-$k$ query aims at finding $k$ instances with the highest (or lowest) scores from multiple lists of instances. Assume there are $m$ parties and $N$ instances, each party $i$ has a score for each instance $j$, denoted by $s_j^i$. On each party, the instances are sorted by their scores. Globally, each instance is assigned an overall score by combining the scores on all parties using an

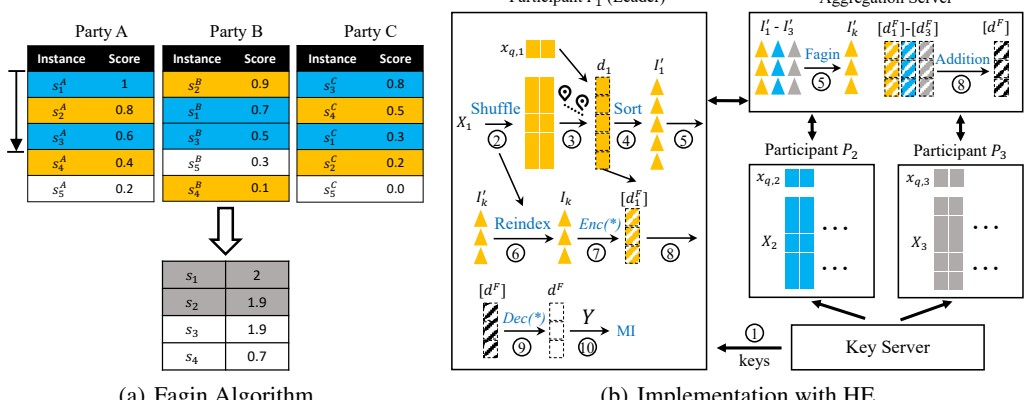

(a) Fagin Algorithm          (b) Implementation with HE

Figure 2: Fagin-inspired method for estimating MI

aggregate function. This problem has been extensively studied by a series of algorithms [31]. In this work, we choose Fagin as it is widely used [18]. Fagin requires that the aggregation function `aggr` is monotone: $\texttt{aggr}(s_1, s_2, ..., s_m) \leq \texttt{aggr}(s'_1, s'_2, ..., s'_m)$ if $s_i < s'_i$ for every $i$.

Figure 2(a) shows an example of Fagin algorithm that selects top-2 instances from three parties with the `addition` aggregation function. The processing of Fagin algorithm contains three major steps: 1) sequentially scan the sorted lists from all parties until there are $k$ instances that have been seen in all lists, 2) obtain the scores of all the instances seen so far (including those have not yet occurred in all lists), and 3) use the aggregation function to compute global scores for candidate instances found in the previous step and select top-$k$ instances therein. In the example of Figure 2(a), Fagin scans three rows and finds that $s_1$ and $s_3$ are in three lists. Afterwards, the scores of all instances in three rows (i.e., $s_1$, $s_2$, $s_3$, $s_4$) are aggregated. We denote $math.Fagin(\cdot)$ the standard Fagin algorithm in this paper, that takes multiple lists of instance IDs as input and outputs candidate IDs.

*Fagin-inspired Optimization for VF-MINE.* We now implement the Fagin-inspired method. The key allocation (step ①) is the same as the baseline method. The key is to aggregate "sub-ranking"s with Fagin algorithm while preserving their privacy. The implementation of the secure `Fagin` operator is shown in Figure 2(b) (step ②-⑤). We propose a shuffle strategy — each participant uses the same seed to randomly shuffle local data instances (step ②). Then, we generate a "pseudo ID" for each instance that is equal to the index after shuffling. Each participant $P_i$ sends the sub-ranking of pseudo IDs, denoted by $I'_i$, to the aggregation server, instead of sending the real IDs (step ③-④). The aggregation server runs the standard Fagin algorithm to find $k$-nearest candidates from all sub-rankings. Since the estimation of MI depends on the label, the leader participant is responsible for stopping Fagin. The aggregation server sends the pseudo IDs of Fagin candidates, denoted by $I'_k$, to each participant (step ⑤). After the `Fagin` operator, the participants re-index these pseudo IDs to their original IDs (step ⑥). The following steps are similar to the baseline method (step ⑦-⑩).

### 3.3    VF-PS: Group Testing-based Participant Selection

The second component in our framework is to search for the *subset* of participants that maximizes MI. We adopt a standard group testing-based search procedure.

*Group-testing Framework.* The VF-PS problem under the group testing framework runs as follows:

1. We select a collection of $T$ *tests*, each of which corresponds to a subset of participants $S_t \subseteq \mathcal{P}$;
2. For each $S_t$, we estimate its mutual information and use it as the *score*;
3. For each participant, we compute its *importance* as the average score in all *tests* it participated in.
4. We pick the top-$k$ most important participants.

Formally, let $U(S)$ be the score of group $S \subseteq \mathcal{P}$ defined as the mutual information over $N_v$ validation data pairs: $V = \{(x^v_j, y^v_j)\}$, and let $\mathbf{A}$ of dimension $T \times m$ be the participating matrix, where $\mathbf{A}_{t,i} = 1$ if participant $i$ belongs to the $t$-th test group. Corresponding to each row of $\mathbf{A}$ is a "score" $U(\mathbf{A}[t]) = U(\{i | \mathbf{A}_{t,i} = 1\})$. Consider $\mathbf{A}$ as the test design, the total score of participant $P_i$ is $u_i = \mathbf{A}[:][i] \cdot [U(\mathbf{A}[0]), ..., U(\mathbf{A}[T-1])]$. The participant selection is conducted using these scores.

---

**Algorithm 1** Fagin-inspired Participant Selection with Communication Batching

---

**Key Server:**

$(pk, sk) = HE.KeyGen()$     //generate homomorphic public and private keys
Send $pk$ to aggregation server, send $(pk, sk)$ to participants

**Participants:**

(Leader) Generate groups $\mathcal{G} = \{S_1, ..., S_T\}$ and send them to aggregation server
  . . . . . .     //find Fagin candidate instances, whose IDs are $I_k$

1: $[d_i^F] = HE.Enc(d_i(I_k); pk)$     //encrypt partial distances of Fagin candidates
2: $Server.addition([d_i^F])$     //send encrypted partial distances to aggregation server
3: $\{[d_S]\} = Server.scheduler()$     //receive aggregated distances from aggregation server
4: $\{I_S'\} = \{math.argsort(HE.Dec([d_S]); sk)\}$     //decrypt and sort aggregated distances
5: $\{I_S\} = Server.scheduler\_callback(\{I_S'\})$     //send local sorted indices to aggregation server
6: (Leader) $scores = group\_testing(\{I_S\}, \mathcal{G}, Y)$     //leader calculates MI scores

**Aggregation Server:**

**function** $Server.addition(\{[d_i^F]\})$:
    $\{[d_{S \in \mathcal{G}}]\} = \{HE.Sum(\{[d_i^F]; p_i \in S\}; pk); S \in \mathcal{G}\}$
**function** $Server.scheduler()$:
    randomly assign $\{[d_{S \in \mathcal{G}}]\}$ to participants
**function** $Server.scheduler\_callback(\{I_S'\})$:
    return $\{I_S\} = reindex(\{I_S'\}; S \in \mathcal{G})$ to leader participant

---

*Implementation of Group Testing.* The implementation of group testing for Fagin-inspired method is quite natural. Specifically, we run Fagin-inspired MI estimation over each test group (subset of participants). The counterpart using the baseline method can be obtained by skipping the Fagin stage.

1. *Generate testing groups.* The leader participant randomly generates $T$ groups (e.g., the first group $S_1 = \{P_1\}$ and the second group $S_2 = \{P_1, P_2\}$), which are sent to other participants.
2. *Run Fagin-inspired method.* The proposed framework handles each test group $t$ independently. Specifically, for each validation instance, the participants in test group $\mathbf{A}[t]$ jointly run Fagin-inspired KNN and calculate the mutual information. The score of the group, $U(\mathbf{A}[t])$, is averaged over all the validation instances.
3. *Participant selection.* Once running all the groups, the leader participant calculates the MI-based score of each participant, with which most valuable participants are selected.

### 3.3.1 Group Testing with Communication Batching

The naive implementation of the group testing routine has a critical drawback. The Fagin candidates generated by different groups may contain overlapping instances, causing significant amount of redundant overheads. As we will show, by leveraging what we call the *Inclusion Property* of the Fagin algorithm, we can batch the communication and computation of all test groups.

*Inclusion Property of Fagin.* The principle of the Fagin algorithm is to scan the sub-rankings from multiple participants until $k$ distinct instances are found in all the sub-rankings. Given two subsets of participants, $\mathcal{S}_1$ and $\mathcal{S}_2$, what we can say about their data access pattern?

**Theorem 1.** *(Fagin's Inclusion Property) Given two groups of participants $\mathcal{S}_1 \subseteq \mathcal{P}$ and $\mathcal{S}_2 \subseteq \mathcal{P}$, their k-nearest candidates generated by Fagin algorithm satisfy $\mathcal{F}_1 \subseteq \mathcal{F}_2$ if $\mathcal{S}_1 \subseteq \mathcal{S}_2$.*

*Proof.* For the group $\mathcal{S}_2$, assume the Fagin algorithm scans $F$ rows of sub-rankings $\{R_i, \forall p_i \in \mathcal{S}_2\}$ before finding instances $D = \{x_1, x_2, ..., x_k\}$ occurring in $\forall p \in \mathcal{S}_2$. The ranking of each instance in $D$ should satisfy:

$$I(x_i, p_j) \leq F, \quad for \ \ \forall x_i \in D \ \ and \ \ p_j \in \mathcal{S}_2$$

Since $\mathcal{S}_1$ is a subset of $\mathcal{S}_2$, we can directly infer:

$$I(x_i, p_k) \leq F, \quad for \ \ \forall x_i \in D \ \ and \ \ p_k \in \mathcal{S}_1$$

The above implies that we can find $D$ in each participant $p \in \mathcal{S}_1$ before scanning $F$ rows The Fagin candidate set of $\mathcal{S}_2$ contains all the unique instances within $F$ rows:

$$\mathcal{F}_2 = Set(\cup_{p_i \in \mathcal{S}_2} R_i[: F])$$

| Datasets | Synthesis | G2-4 | G2-128 | Bank | Unbalance | Letter | Birch1 | Birch2 |
|---|---|---|---|---|---|---|---|---|
| # instances | 1000 | 2048 | 2048 | 3200 | 6500 | 20K | 100K | 100K |
| # features | 50 | 4 | 128 | 8 | 2 | 16 | 2 | 2 |
| # classes | 2 | 2 | 2 | 2 | 8 | 26 | 100 | 100 |
| # partitions | 5 | 4 | 4 | 4 | 2 | 4 | 2 | 2 |

Table 1: Evaluated datasets.

Similarly, the Fagin candidate set of $\mathcal{S}_1$ is a subset of all unique samples within $F$ rows:

$$\mathcal{F}_1 \subseteq Set(\cup_{p_i \in \mathcal{S}_1} R_i[: F])$$

Since $\mathcal{S}_1$ is a subset of $\mathcal{S}_2$, all the samples appear within $F$ rows of $\mathcal{S}_1$ also appear in $\mathcal{S}_2$:

$$Set(\cup_{p_i \in \mathcal{S}_1} R_i[: F]) \subseteq Set(\cup_{p_i \in \mathcal{S}_2} R_i[: F])$$

Therefore, we can conclude: $\mathcal{F}_1 \subseteq \mathcal{F}_2$.

$\square$

Theorem 1 indicates that the Fagin candidates of all groups can be found in those of the *"complete group"* that consists of all the participants. Based on this observation, we propose to batch the Fagin tasks of different groups in a single execution.

*Implementation of Batch Optimization.* The batching optimization is illustrated in Algorithm 1. The leader participant generates groups $\mathcal{G} = \{S_1, ...S_T\}$ and sends $\mathcal{G}$ to the aggregation server. The following shows the steps after getting the Fagin candidates, $I_k$, for the complete group $\mathcal{P}$.

1. Each participant $P_i$ encrypts partial distances with homomorphic encryption. (line 1)
2. Each participant sends encrypted partial distances of Fagin candidates, i.e., $[\mathbf{d}_i^F]$, to the aggregation server through $Server.addition$ (line 2). For each group $S \in \mathcal{G}$, the aggregation server calculates the sum of all partial distances in $S$.
3. A random scheduler, $Server.scheduler$, on the aggregation server sends the aggregated distances of each group (denoted by $[\mathbf{d}_S]$) to a participant randomly (line 3).
4. For each received $[\mathbf{d}_S]$, the participant performs decryption and gets the sorted index $I'_S$ (line 4).
5. The sorted indices are sent to the aggregation server through $Server.scheduler\_callback$ (line 5). The aggregation server lets the random scheduler associate them with the corresponding groups. Then, the aggregation server sends the sorted IDs of all the groups, $\{I_S\}$, to the leader participant.
6. The leader participant calculates the MI-based scores using the method of group testing and performs participant selection (line 6).

### 3.4 Security Analysis

In this work, we assume that the key server is honest and does not collude with any involved party. The aggregation server and all the participants are honest-but-curious, a common knowledge threat model used in the federated learning literature [28, 32]. The security properties of our proposed methods are summarized as follows (recall security requirements in Section 3.1). *Feature security* is protected against any curious party since the participants do not share their local features. *Aggregation security* is assured against curious aggregation server by homomorphic encryption. Regarding curious or colluding participants, aggregation security is also assured if the leader participant is benign. *Identity security* is achieved against a curious server if there is no server-participant collusion, because the proposed methods do not share instance identifiers $I$ with the server.

## 4 Evaluation

We conducted experiments to validate the efficiency and effectiveness of the proposed methods.

### 4.1 Experimental Setting

*Implementation.* Numpy and PyTorch libraries are used to perform data loading and tensor manipulations. We implement RPC communication with proto2 and gRPC. TenSEAL [33] is a homomorphic library built on top of Microsoft SEAL [34]. We run all experiments in a cluster, in which each machine is equipped with 24 GB memory, 8 cores and 10 Gbps bandwidth.

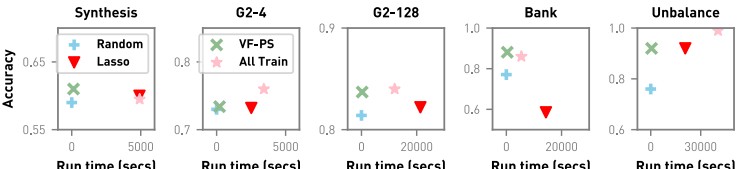

Figure 3: Selection performance (Pivot, 50% participants). "All-Train" refers to training on all participants. The other datasets are not reported because each training task costs more than one day.

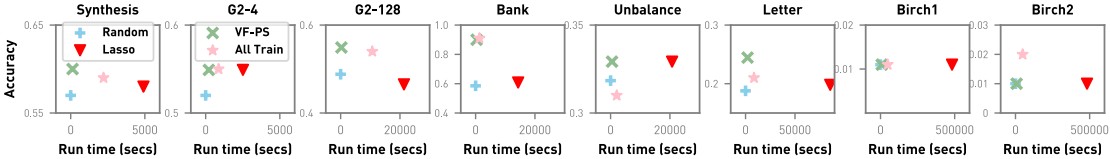

Figure 4: Selection performance (LR, 50% participants).

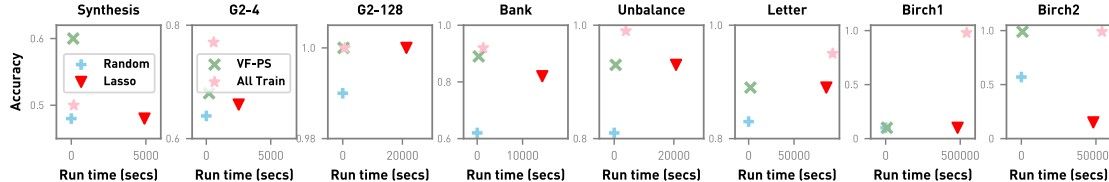

Figure 5: Selection performance (KNN, 50% participants).

_Datasets._ Table 1 illustrates all datasets that we used. Synthesis is a synthesis dataset by randomly generating a separation hyperplane, based on which data instances are sampled. The other datasets are collected from online repository [35, 36] and prior work [37–39]. Each dataset is split into a training set (80%), a validation set (10%), and a test set (10%). We randomly partition the training set into several vertical partitions and put each partition on one machine.

_ML Models._ We choose Pivot [9], a vertically federated tree training engine, as the downstream ML model. Additionally, we also implement _logistic regression_ (LR), and _k-nearest neighbor_ (KNN).

_Participant Selection Baselines._ We compare the following baselines on participant selection. 1) RANDOM: We randomly choose $l$ participants and run ML model. The result is averaged over ten runs to assure the robustness. 2) LASSO. Lasso regression is a standard feature selection method in _scikit-learn_. We train a Lasso model and choose high-importance participants according to the absolute sum of coefficients on each participant. 3) VF-PS is our proposed method.

_Protocols._ We implement Adam [40] and vanilla SGD as the optimization algorithm for LR and Lasso. We grid search the optimal learning rate in $\{0.001, 0.01, 0.1\}$ and the regularization term in $\{10^{-4}, 10^{-3}, 10^{-2}\}$, and set batch size to 256. We terminate the task after 50 epochs or the validation loss does not decrease within 5 epochs. For Pivot, we choose GBDT tree, and tune tree depth in $\{2, 3, 4\}$ and tree number in $\{1, 2, 3\}$. The number of groups $T$ is set to 10. The other hyper-parameters are set as default.

## 4.2 Evaluation of Participant Selection

_Selection Performance._ We first study the following question: _can our proposed method outperform baselines regarding the selection of high-importance participants?_ We compare RANDOM, LASSO and VF-PS in terms of their performance of participant selection. When the downstream classification model is Pivot, Figure 3 shows the performance of all methods for selecting 50% participants. RANDOM is the fastest by instantly choosing participants; however, its selection accuracy can be much worse on many datasets. LASSO is the slowest since it needs to run multiple epochs until convergence and tunes the hyper-parameters. VF-PS is only slightly slower than RANDOM and significantly faster than LASSO. Meanwhile, VF-PS selects comparable, and often better, subsets of participants, indicated by the accuracy of training Pivot only on selected subsets. When the classification model becomes LR and KNN, we observe similar phenomena, as shown in Figure 4-5.

_Comparison to Full-fledged Training._ We then turn to another question: _does participant selection provide benefits compared to training with all the participants?_ To answer this question, we also

|             | Synthesis | G2-4 | G2-128 | Bank | Unbalance | Letter | Birch1 | Birch2 |
|-------------|-----------|------|--------|------|-----------|--------|--------|--------|
| VF-Ps/50%   | 0.60      | 0.68 | 1.0    | 0.89 | 0.93      | 0.89   | 0.1    | 0.99   |
| VF-Ps/25%   | 0.59      | 0.61 | 1.0    | 0.8  | -         | 0.83   | -      | -      |
| Brute-force/50% | 0.60  | 0.68 | 1.0    | 0.93 | 0.93      | 0.94   | 0.1    | 0.99   |
| Brute-force/25% | 0.59  | 0.63 | 1.0    | 0.93 | -         | 0.86   | -      | -      |

Table 2: Comparison to the brute-force strategy on KNN. The metric is the test accuracy.

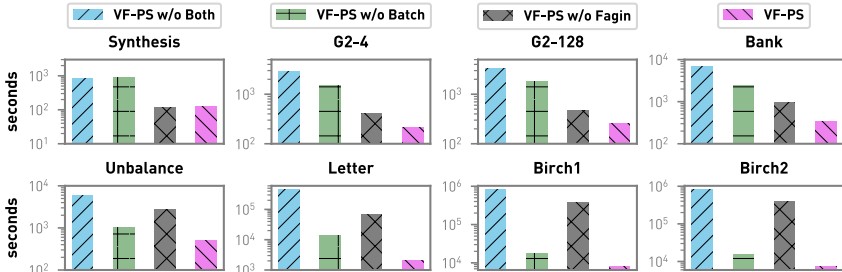

Figure 6: Ablation study of our proposed method.

show the result of training using all participants ("All-Train") in Figure 3-5. As we can see, the selection time of VF-Ps is orders of magnitude faster than the training time using all the participants; meanwhile, if we were to train a model only on the selected subset, we can achieve similar test accuracy on most datasets. On some datasets, e.g., Synthesis, G2-128, and Letter, we even observe higher accuracy after participant selection compared to the full-fledged training. This shows that some participants may have negative impacts on the model quality, and it is beneficial to choose high-importance participants. These experimental results verify the effectiveness and efficiency of participant selection.

*Comparison to Brute-force Strategy.* Intuitively, the brute-force solution for participant selection is to enumerate all the possible groups and choose the best one. Although this method is too time-consuming (up to two orders of magnitude slower according to our results), it provides the best possible selection strategy that can be used to understand — *how big is the gap between our selection strategy and the optimal (brute-force strategy)?* Table 2 shows the results of VF-Ps and brute-force on KNN (other models are too slow to run the brute-force strategy). When the selection ratio is 50%, VF-Ps gets the same selection quality on six datasets out of eight. Overall, the gap of selection quality is moderate on most datasets. It is an interesting future direction to come up with better participant selection criteria, going beyond mutual information.

*Ablation Study.* We now validate that both system optimizations (Fagin and Batching) significantly improve the system performance. We compare VF-Ps with three variants—w/o Fagin, w/o batching, and w/o both—as illustrated in Figure 6. We see that both optimizations are crucial to the performance — disabling either of them significantly slows down the runtime on all datasets.

## 5   Related Work

*Federated Learning.* Federated Learning (FL) is a category of distributed machine learning approach that lets multiple clients jointly train machine learning models over their personal data without data leakage [3–5]. Since the feature space and sample space of participating parties may not be identical, federated learning can be classified into horizontally federated learning and vertically federated learning. Horizontally federated learning is a scenario in which different parties share the same feature space but different sample space. In contrast, in vertically federated learning, different parties have the same sample space but different feature space.

*Vertically Federated Learning.* Hardy et al. [8] proposed a three-party vertically federated logistic regression solution by entity resolution and additively homomorphic encryption. Yang et al. [41] from WeBank and Yang et al. [42] from Baidu further extended vertically federated logistic regression to Quasi-Newton optimization and non-coordinator scenario. Feng and Yu [6] designed a multi-participant and multi-class federated learning framework for vertical training data. VAFL [7] was a vertically federated learning framework that allows each client to asynchronously run stochastic gradient algorithms without coordination with other clients. Pyvertical [43] was proposed to train multi-headed SplitNNs in the context of vertically federated learning using private set intersection. Liu et al. [44] proposed an asymmetrical vertically federated learning framework that protects sample

IDs with the private set intersection protocol. Pivot [9] was proposed to train privacy-preserving vertical decision tree using MPC protocol.

*Federated Participant Selection.* In the horizontally federated scenario, FedCS [45] tried to solve the resource heterogeneity in federated setting and conducted client selection to accelerate system performance. FedMCCS [46] further introduced a multicriteria approach for client selection, considering clients resources including CPU, memory, energy, and time. To address limited bandwidth in the presence of many clients, Huang et al. [47] proposed a selection policy to improve training efficiency while assuring client fairness. Different from these works that studied participant selection for system heterogeneity [48], our work focuses on selection regarding the contribution of each individual participant for the trained model. A few early, seminal works studied contribution-based participant selection in horizontally federated learning [49].

## 6 Limitation

Although VF-PS works well on a range of workloads, it still has several limitations:

- *Multiple training stages.* In this paper, we treat VF-PS as a task that is similar to feature selection, which is a pre-processing step that is often "once-and-for-all". However, there are use cases in which the importance of each participant changes during different training stages, then VF-PS may not find the optimal result.
- *Reward assessment.* The goal of this work is to measure the mutual information of each participant. To come up with a "reward", it needs to be carefully "normalized" over the "importance" of each participant — directly using the mutual information provided by VF-PS as "reward" could cause problems on fairness. This problem definitely requires careful further studies.
- *Generalization of MI estimator.* KNN-based MI estimator can be biased towards smaller local feature subsets with Euclidean distance. Our technique can also go beyond Euclidean distance — as long as local scores can be aggregated using a monotonic aggregation function, most, if not all, of our optimizations can still be applied. But still, whatever limitations that MI-based method has for feature selection, our methods will probably also inherit. We will study other distance metrics in future work.
- *Large-scale federated network.* It is an interesting future research direction to understand real-world VFL scenarios where the federated network consists of an even larger number of participants/organizations. When this number becomes very large, we do expect that VF-PS could encounter performance degradation due to the limitation of the Fagin strategy. We will study this challenging problem in future work.

## 7 Conclusion

In this work, we study participant selection in the context of vertically federated learning. We model this problem as a mutual information-based view. We first propose a novel mutual information estimator, called VF-MINE, that uses Fagin's algorithm. To perform participant selection, we propose a group testing-base framework VF-PS on top of VF-MINE. We implement our proposed method using homomorphic encryption and design a batching optimization. We analyze the security properties of the proposed methods and show that our methods achieve orders of magnitude improvements.

## Acknowledgments and Disclosure of Funding

This work is supported by the National Natural Science Foundation of China under Grant No. 62225113, OceanBase, and Ant Group. CZ and the DS3Lab gratefully acknowledge the support from the Swiss State Secretariat for Education, Research and Innovation (SERI) under contract number MB22.00036 (for European Research Council (ERC) Starting Grant TRIDENT 101042665), the Swiss National Science Foundation (Project Number 200021_184628, and 197485), Innosuisse/SNF BRIDGE Discovery (Project Number 40B2-0_187132), European Union Horizon 2020 Research and Innovation Programme (DAPHNE, 957407), Botnar Research Centre for Child Health, Swiss Data Science Center, Alibaba, Cisco, eBay, Google Focused Research Awards, Kuaishou Inc., Oracle Labs, Zurich Insurance, and the Department of Computer Science at ETH Zurich.

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
