# OpenReview forum: "VF-PS: How to Select Important Participants in Vertical Federated Learning, Efficiently and Securely?"
_NeurIPS.cc/2022/Conference — NeurIPS 2022 Accept_

### Official Review · Reviewer_jzaC · 2022-06-29

**Rating:** 6
**Confidence:** 3
**Soundness:** 3 good
**Presentation:** 3 good
**Contribution:** 3 good

**Summary:**

The paper studies the problem of client sampling in the context of Vertical FL, i.e., FL-PS, where each client corresponds to a feature subset and thus differs in their importance to the global model. Specifically, a mutual information (MI) based approach is used for importance evaluation. The Fagin algorithm and a so-called batched group testing scheme are employed to make the selection efficient. Security analysis is provided. Results of evaluation on various classification datasets show that the proposed framework is competitve to full-participation training with very low sampling overheads.

**Questions:**

- Is it possible that the kNN-based MI estimator is biased towards smaller local feature subsets with Euclidean distance?
- In the Security Analysis, is the framework still secure when the leader participant is curious and seeks to reconstruct non-local features?
- How are the groups for testing generated? How is the number of groups $T$ determined?

**Limitations:**

The methodology, in my understanding, is practically limited from two aspects: i) a globally aligned, complete instance space $\mathcal{D}$, as mentioned above, and that ii) the MI-based design is only applicable to classification tasks, which is a common limitation for most VFL algorithms/frameworks though.

**Strengths And Weaknesses:**

Strengths:
- Client/feature subsets sampling in VFL is an intriguing problem under-explored in the field of FL.
- The authors introduce multiple techniques, e.g., Fagin, group testing, batching, to make their MI-based sampling framework more practicaly feasible.
- The empirical results are relatively comprehensive, covering different scale of data settings and multiple models.
- Privacy protection is considered in the whole process.

Weaknesses:
- An identical instance space $\mathcal{D}$ is assumed, which may not be the case for many VFL scenarios.
- The complexity  and efficacy of group testing scheme is subject to how groups are generated. Random grouping over a multitude of clients can be problematic.
- Some usage of symbols and terms seem to be too casual. E.g., "# participants", "$NN_{k, y_q}$", "All-train", etc.

---

> ### Author Response · Authors · 2022-08-02
> **Response to Reviewer jzaC**
>
> __Q1: Is it possible that the kNN-based MI estimator is biased towards smaller local feature subsets with Euclidean distance?__
>
> Response: Thanks for this insight! Indeed, with Euclidean distance, this can definitely happen, and we will make this limitation clear in the paper.
>
> Our technique can also go beyond Euclidean distance —  as long as local scores can be aggregated using a monotonic aggregation function, most, if not all, of our optimizations can still be applied. Unfortunately, we do not have time to finish running all experiments during the rebuttal period, but we will add several new distances in the revised version of this paper.
>
> But still, whatever limitations that MI-based method has for feature selection, our methods will probably also inherit. We have added a discussion on this limitation in Appendix E of the revised paper.
>
> __Q2: In the Security Analysis, is the framework still secure when the leader participant is curious and seeks to reconstruct non-local features?__
>
> Response: A curious leader participant does not directly receive [di] from other participants, and hence cannot speculate benign participants. However, if the leader participant further colludes with some participants, it may speculate other participants’ partial distances and reconstruct their features. We leave the detailed discussions to Appendix B.
>
> __Q3: How are the groups for testing generated? How is the number of groups T determined?__
>
> Response: We generate all possible groups and randomly choose T groups. Intuitively, the value of T can affect the selection performance. We have added experiments in Appendix D4 of the revised paper to assess the impact of T. As shown in Figure 11, the accuracy of selection results increases with a larger T. This is unsurprising because adding more groups can try more group combinations and therefore improve the selection quality. This was also verified by previous works, e.g., Theorem 2.2 and Figure 5 in [1]. Besides, the selection time also increases with more test groups, but the increase is not significant due to our batching mechanism. In our experiments, we set T to 10 since it achieves a good trade-off between accuracy and selection cost.
>
> [1] Yingbo Zhou, Utkarsh Porwal, Ce Zhang, Hung Q Ngo, XuanLong Nguyen, Christopher Re, and Venu Govindaraju. Parallel feature selection inspired by group testing. Advances in Neural Information Processing Systems, 2014.

---

### Official Review · Reviewer_63Y2 · 2022-07-07

**Rating:** 6
**Confidence:** 3
**Ethics Flag:** Yes
**Soundness:** 3 good
**Presentation:** 3 good
**Contribution:** 3 good

**Summary:**

The authors studies the problem of participant selection in vertical federated learning, and model it with a mutual information-based view. The first technical contribution is VF-MINE, a vertically federated mutual information estimator. The second contribution is to further optimize VF-MINE and get a group testing-based participant selection framework, VF-PS. Some experimental evaluations were conducted to show the performance of VF-PS.

**Questions:**

The paper seems to suggest that VS-PF is a once-and-for-all method. But what if the “good selection of participants” varies in different stages of training?

**Ethics Review Area:**

["Discrimination / Bias / Fairness Concerns"]

**Limitations:**

The authors have addressed the limitations that the mutual information may not be the best selection criterion. Beyond that, the authors may need to consider the social impact of inequality of contribution and reward between selected and unselected participants.

**Strengths And Weaknesses:**

**Strengths:**
+ The idea of boosting the efficiency of vertical federated learning with participant selection is novel and interesting.
+ The Fagin-inspired optimization is novel.
+ The analysis of security and the experimental results are good.


**Weaknesses:**
- Some parts of the algorithms are not clear and easy to follow. Please see my questions below.

---

> ### Author Response · Authors · 2022-08-02
> **Response to Reviewer 63Y2**
>
> __Q1: The paper seems to suggest that VF-PS is a once-and-for-all method. But what if the “good selection of participants” varies in different stages of training?__
>
> Response: In this paper, we treat VF-PS as a task that is similar to feature selection, which is a pre-processing step that is often “once-and-for-all”. However, the reviewer makes a great point — we can definitely imagine use cases in which the importance of each participant changes during different stages of training — we have added a discussion on this potential limitation in Appendix E of the revised paper.
>
> __Q2: The authors may need to consider the social impact of inequality of contribution and reward between selected and unselected participants.__
>
> Response: We definitely agree! The goal of this work is to measure the mutual information of each participant. To come up with a “reward”, it needs to be carefully “normalized” over the “importance” of each participant — directly using the mutual information provided by VF-PS as “reward” could cause problems on fairness. This problem definitely requires careful further studies; in the revised version, we added a discussion on this problem in Appendix E.

---

### Official Review · Reviewer_2byC · 2022-07-12

**Rating:** 7
**Confidence:** 3
**Soundness:** 3 good
**Presentation:** 3 good
**Contribution:** 3 good

**Summary:**

In this work, the authors propose a new method for efficient vertical FL. Specifically, the authors propose to find a small subset of clients whose features could be used to learn a model that yields high accuracy with significantly less amount of time using 1. Mutual Information to evaluate how informative a selected subset of features is, 2. a group testing method. Empirical results show that the proposed method is able to achieve comparable/stronger performance than regular training with significantly less time.

**Questions:**

Questions are listed in the previous section.

**Limitations:**

Limitations are listed in weakness section.

**Strengths And Weaknesses:**

Strength:
- Overall this paper is well written with clear intuitions on how each component of the algorithm is designed. The novel idea of using mutual information with homomorphic encryption is interesting and sensible.
- The authors evaluate on a thorough set of datasets and the empirical results show that the proposed method is strong compared to prior baselines.

Weakness:
- The authors propose a group testing method to select the best k clients. However, this method could be potentially costly. Could the authors provide any justification how the number of tests $T$ affects the convergence of finding best top-k clients? Moreover, the experiments in this work focus on cases where the number of clients is small (<5). Could this method generalize to large scale federated network where the number of partitions is large?
- For all the datasets evaluated in this work, the authors use a logistic regression model to evaluate the proposed method. Could the authors provide evaluations of VF-PS on more sophisticated ML task, e.g. non convex loss?

---

> ### Author Response · Authors · 2022-08-02
> **Response to Reviewer 2byC**
>
> __Q1: Could the authors provide any justification how the number of tests T affects the convergence of finding best top-k clients?__
>
> Response: Thanks for this great suggestion! Intuitively, the value of T can affect the selection performance. We have added experiments in Appendix D4 of the revised paper to assess the impact of T. As shown in Figure 11, the accuracy of selection results increases with a larger T. This is unsurprising because adding more groups can try more group combinations and therefore improve the selection quality. This was also verified by previous works, e.g., Theorem 2.2 and Figure 5 in [1]. Besides, the selection time also increases with more test groups, but the increase is not significant due to our batching mechanism. Moreover, for larger T, VF-PS is still significantly faster than other methods (we have not finished running all baselines since they are too slow; but we will add them to Figure 11 in the final version).
>
> [1] Yingbo Zhou, Utkarsh Porwal, Ce Zhang, Hung Q Ngo, XuanLong Nguyen, Christopher Re, and Venu Govindaraju. Parallel feature selection inspired by group testing. Advances in Neural Information Processing Systems, 2014.
>
>
> __Q2: Could this method generalize to large scale federated network where the number of partitions is large?__
>
> Response: This is a great question! Unlike the horizontally federated settings, most VFL settings have a relatively small number of participants (e.g., most recent papers [2-6] only evaluate up to 10 participants). In Appendix D3, we show that up to 16 participants, VF-PS significantly outperforms other methods with the batching optimization.
>
> It is definitely a very interesting future research direction to understand real-world VFL scenarios where the federated network consists of an even larger number of participants/organizations! When this number becomes very large, we do expect that VF-PS could encounter performance degradation because the Fagin strategy becomes less effective for a large group (Appendix D3 also proves this). We have revised Appendix E to include a paragraph of discussion about this.
>
>
> [2] Shengwen Yang, Bing Ren, Xuhui Zhou, and Liping Liu. Parallel distributed logistic regression for vertical federated learning without third-party coordinator. arXiv:1911.09824, 2019.\
> [3] Tianyi Chen, Xiao Jin, Yuejiao Sun, and Wotao Yin. Vafl: a method of vertical asynchronous federated learning. arXiv:2007.06081, 2020.\
> [4] Yang Liu, Xiong Zhang, and Libin Wang. Asymmetrical vertical federated learning. arXiv:2004.07427, 2020.\
> [5] Kai Yang, Tao Fan, Tianjian Chen, Yuanming Shi, and Qiang Yang. A quasi-newton method based vertical federated learning framework for logistic regression. arXiv:1912.00513, 2019.\
> [6] Yuncheng Wu, Shaofeng Cai, Xiaokui Xiao, Gang Chen, and Beng Chin Ooi. Privacy preserving vertical federated learning for tree-based models. Proceedings of the VLDB Endowment, 2020.\
>
>
> __Q3: Could the authors provide evaluations of VF-PS on more sophisticated ML task, e.g. non convex loss?__
>
> Response: We agree that it is interesting to show how VF-PS performs over more sophisticated ML tasks. (In the original manuscript, we ran the GBDT model in Pivot which has a non-convex loss (Figure 3))
>
> To further strengthen the evaluation, we have added an experiment with neural network models. We run the vertically federated selection method and then train a neural network model over selected features. The evaluated model is an MLP (multilayer perceptron) that has two fully-connected layers. The hidden size is 32 and the batch size is 32. In Appendix D2, Table 3 presents the results of three baselines over Letter dataset. VF-PS achieves the best selection performance for choosing 25% and 50% of participants. Specifically, the test accuracy trained with the entire dataset is 0.6, and VF-PS can obtain an accuracy of 0.58 after selecting 50% participants.
>
> Note that, in this experiment, we train the neural network model in a centralized way because we cannot find any practical open-source framework (any suggestion would also be great!). We will include neural network models into our framework if practical implementations under the vertically federated setting become available in the future.

---

### Review · Ethics_Reviewer_Jto8 · 2022-08-02

**Recommendation:**

 I would recommend the authors make space in the main paper for a short discussion that includes at a minimum an example of how this could be problematic and references to resources for how to detect and address fairness problems like representation bias.

Mehrabi, Ninareh, et al. "A survey on bias and fairness in machine learning." ACM Computing Surveys (CSUR) 54.6 (2021): 1-35.

Suresh, Harini, and John Guttag. "Understanding Potential Sources of Harm throughout the Machine Learning Life Cycle." (2021)


**Ethical Issues:**

Yes

**Ethics Review:**


Yes, this paper’s methodology raises concerns about downstream negative societal impacts if the set of participants selected disproportionately represented certain subgroups of the population at the expense (or benefit) to others.

---

### Review · Ethics_Reviewer_B2Xp · 2022-08-02

**Recommendation:** See the ethics review above.

**Ethical Issues:**

Yes

**Ethics Review:**

In line with reviewer 63Y2 (and now, after the rebuttal, also the authors), I agree that the method could create unfairness as only a subset of participants are selected. The authors addressed these concerns to some degree in the appendix but I would like to see two things:
1) NeurIPS offered an extra page this year compared to last year, precisely to make room for addressing things which are mentioned in the checklist. This includes limitations and broader impacts so these should be in the main manuscript. Note that the camera ready version offers an extra page which could you be used for this purpose.
2) The authors should go into more depth around what kind of unfairness could emerge and what impact this could have. The limitation section should help practitioners or other researchers that build on their work.

---

### Author Response · Authors · 2022-08-02
**Common Responses to All Reviewers**

We are grateful to the reviewers for their careful reviews and suggestive comments. Following the comments, we have carefully revised and improved our paper at full stretch. The changes made in the revised manuscript are highlighted in **blue**. Below, we present our responses to the comments from each reviewer individually.

---

### Meta-Review · Area_Chair_oEWx · 2022-08-29

**Recommendation:** Accept
**Confidence:** Less certain

**Metareview:**

This paper studies the problem of vertical federated learning, where each client (e.g., hospital) has access to a disjoint subset of features. The reviewers agree that the paper provides an interesting technique for client selection. The authors have adequately addressed the reviewers' questions.

The authors should consider dedicating part of their revision to addressing concerns raised by the ethics reviewers. In particular, there are concerns of unfairness due to the selection of a small subset of participants. Even though the paper does not readily provide a solution to this problem, the authors should acknowledge this potential issue and suggest directions for future work. If there is not enough space in the paper, the authors could use part of the appendix for a detailed discussion.

**Award:**

No

---

### Decision · Program_Chairs · 2022-09-14

Accept